# Linear and Nonlinear Land Use Regression Approach for Modelling PM$_{2.5}$ Concentration in Ulaanbaatar, Mongolia during Peak Hours

Odbaatar Enkhjargal [1,2], Munkhnasan Lamchin [3,4,*], Jonathan Chambers [5] and Xue-Yi You [1]

[1] Department of Environmental Science, School of Environmental Science and Engineering, Tianjin University, Tianjin 300350, China
[2] Division of Physical Geography and Environmental Study, Institute of Geography and Geoecology, Mongolian Academy of Sciences, Ulaanbaatar 15170, Mongolia
[3] Ojeong Eco-Resilience Institute (OJERI), Korea University, 145, Anam-Ro, Seongbuk Gu, Seoul 02841, Republic of Korea
[4] Department of Environment and Forest Engineering, School of Engineering and Applied Scences, Institute for Sustainable Development, National University of Mongolia, Ulaanbaatar 14201, Mongolia
[5] United Nations Organization Stabilization Mission in the Democratic Republic of the Congo (MONUSCO), 12 Avenue des Aviateurs, Gombe, Kinshasa BP 8811, Congo
* Correspondence: nasaa@korea.ac.kr

**Abstract:** In recent decades, air pollution in Ulaanbaatar has become a challenge regarding the health of the citizens of Ulaanbaatar, due to coal combustion in the ger area. Households burn fuel for cooking and to warm their houses in the morning and evening. This creates a difference between daytime and nighttime air pollution levels. The accurate mapping of air pollution and assessment of exposure to air pollution have thus become important study objects for researchers. The city center is where most air quality monitoring stations are located, but they are unable to monitor every residential region, particularly the ger area, which is where most particulate matter pollution originates. Due to this circumstance, it is difficult to construct an LUR model for the entire capital city's residential region. This study aims to map peak PM$_{2.5}$ dispersion during the day using the Linear and Nonlinear Land Use Regression (LUR) model (Multi-Linear Regression Model (MLRM) and Generalized Additive Model (GAM)) for Ulaanbaatar, with monitoring station measurements and mobile device (DUST TRUK II) measurements. LUR models are frequently used to map small-scale spatial variations in element levels for various types of air pollution, based on measurements and geographical predictors. PM$_{2.5}$ measurement data were collected and analyzed in the R statistical software and ArcGIS. The results showed the dispersion map MLRM $R^2 = 0.84$, adjusted $R^2 = 0.83$, RMSE = 53.25 µg/m$^3$ and GAM $R^2 = 0.89$, and adjusted $R^2 = 0.87$, RMSE = 44 µg/m$^3$. In order to validate the models, the LOOCV technique was run on both the MLRM and GAM. Their performance was also high, with LOOCV $R^2 = 0.83$, RMSE = 55.6 µg/m$^3$, MAE = 38.7 µg/m$^3$, and GAM LOOCV $R^2 = 0.77$, RMSE = 65.5 µg/m$^3$, MAE = 47.7 µg/m$^3$. From these results, the LUR model's performance is high, especially the GAM model, which works better than MRLM.

**Keywords:** fine particulate matter; Linear and Nonlinear LUR model; air pollution; Ulaanbaatar

## 1. Introduction

Air pollution is widely recognized as a significant public health risk [1], particularly fine particulate matter (PM$_{2.5}$), which travels deep within the respiratory tract, eventually reaching the lungs [2]. This type of particulate matter air pollution can affect our health in many ways, with both short-term and long-term effects [3]. The short-term effects can include irritation to the eyes, nose, and throat, as well as upper respiratory infections such as bronchitis and pneumonia. The long-term effects can include chronic respiratory disease, lung cancer, heart diseases, and even damage to the brain, nerves, liver, or kidneys [4].

Worldwide, 4.2 million deaths every year occur as a result of exposure to ambient air pollution [5].

Ulaanbaatar, the capital city of Mongolia, is the coldest capital city on Earth. Its average ambient temperatures routinely fall below $-40\ ^\circ$C between November and February [6]. In such cold temperatures, keeping warm is a significant issue for the city's inhabitants, where the central heating system is not well developed. To survive, citizens burn coal and wood during the winter months. Consequently, the ger area has become the city's primary source of air pollution [1]. A ger is a typical Mongolian traditional house built of felt and wood. A sizeable population resides in the ger neighborhood, which is located on the outskirts of Ulaanbaatar. This type of rapid urbanization has led to deteriorating air quality in the city, with emissions originating from domestic heating and cooking needs, an increasing number of vehicles, a growing number of industries, construction activities, and higher electricity demands [4]. As of 2020, 1,597,300 people were living in Ulaanbaatar, or 47.5% of the total population of Mongolia [7]. Of these, 391,698 households, or 51.3% percent, were living in the ger area, using a stove as a home heating device [8]. The Ger area's main heating sources comprise various stoves, such as kilns, traditional Mongolian metal stoves, heat-only boilers, improved stoves, and a small number of electric heating devices. The ger area mainly consists of two types of housing, gers (Mongolian traditional dwelling) and houses, both of which are heated by stoves. On average, gers are estimated to consume 3.8 tons of coal and 2.4 m$^3$ of fuelwood per year, and houses are estimated to consume 4.3 tons of coal and 2.4 m$^3$ of fuelwood per year [9]. According to IQ Air (World Air Quality) research, Ulaanbaatar was ranked as the third most polluted capital city in the world in 2020 [10], and as of 31 January 2022, Ulaanbaatar was ranked fourth in the Live Air Quality city ranking [11].

To decrease air pollution, the Mongolian government has implemented several projects, such as the distribution of improved stoves (since 2012) and the use of "Tavan Tolgoi Briquettes" (since 2019) instead of raw coal. The "improved stove" program ended in 2013, and as of 2021, in the ger area, 55% of householders were using improved stoves, while the rest were using traditional stoves [9]. According to an experiment by Yukimasa Takemoto et al. on stove chimney emissions [12], traditional stoves emit 3.2 g of soot particles and improved stoves emit 0.6 g of soot particles when coal combustion is 1 kg. The "Tavan Tolgoi Briquette" project is still ongoing. Even though these projects decrease particulate matter pollution at the appropriate level, fine particulate matter air pollution in Ulaanbaatar is still high. In the week of 17–23 January 2022, in Ulaanbaatar, the average PM$_{2.5}$ in a 24 h period peaked at 413 μg/m$^3$ [13], which is 8.2 times higher than the "Air Quality General Requirement" levels specified in MNS4585:2016 [14] and 27.5 times higher than the "WHO Air Quality Guideline: Particulate Matter" levels [15]. However, these are average measurements over 24 h. Additionally, outdoor PM$_{2.5}$ levels vary diurnally, as well as by season, being higher in the morning and the evening than during the rest of the day, since this is when coal burning for domestic heating and incidentally for cooking takes place [6]. In other words, PM$_{2.5}$ pollution peaks early in the morning, from approximately 05:00 to 10:00 a.m., and in the evening from approximately 06:00 to 00:00 p.m. [16]. Generally, it continues for 6 h in the morning and evening, or approximately half of the day, at which time fine particulate matter is very high. For instance, the highest ever fine particulate matter pollution—3320 μg/m$^3$—was recorded at 05.00 a.m. on 31 January 2018 [17]. This over-pollution of PM$_{2.5}$ has been influencing citizens' health for years. In the last three years, the National Statistics Office has published reports regarding epidemiological studies related to air pollution. According to these reports [18,19], per 10,000 people, respiratory disease increased continuously from 956 to 1961 from 2010 to 2019. Children under 5 are especially becoming sick more often. Per 10,000 children under 5, there was an increase from 4036 to 6405 cases from 2010 to 2019 in Ulaanbaatar.

Previous studies [1,4,20–22] have suggested numerous methods for mapping PM$_{2.5}$ in Ulaanbaatar. The scale of the resulting dispersion maps, however, is too great, and some of the maps merely show the city's core, without expanding outward. Some researchers have

used mobile devices for the LUR model due to a lack of dedicated air quality monitoring stations, and a recent review of air pollution dispersion techniques by Xing zhe et al. [23] found that the LUR model was the most widely used method for accurately mapping the PM$_{2.5}$ distribution. The main target of this study is to cover all residential areas of Ulaanbaatar and to define the highest-risk areas by showing the PM$_{2.5}$ gradient accurately with more acceptable techniques.

## 2. Materials and Methodology

### 2.1. Study Area and Sampling

As mentioned above, Ulaanbaatar's PM$_{2.5}$ pollution is far too high early in the morning and in the evening, and this influences Ulaanbaatar citizens' health. Thus, there is a need to map PM$_{2.5}$ peak dispersion accurately and to understand the risk level of the air that we breathe in the morning and evening. Even though Ulaanbaatar's air pollution is measured at 16 locations [24], until now, accurate dispersion mapping methods have not been developed due to the scarcity of monitoring stations, which means that air pollution dispersion is being mapped by dots. Of these stations, the Department of Capital City Air Quality possesses 5, and the other 11 belong to the National Agency of Meteorology and Environmental Monitoring (NAMEM). Some monitoring stations do not measure all pollutants, and only 10 stations measure the PM$_{2.5}$ every 15 min–1 h.

The study area covers all residential areas of Ulaanbaatar, including the outskirts of the city, which amounts to 1464.95 square kilometers (37 km × 39.7 km) (Figure 1).

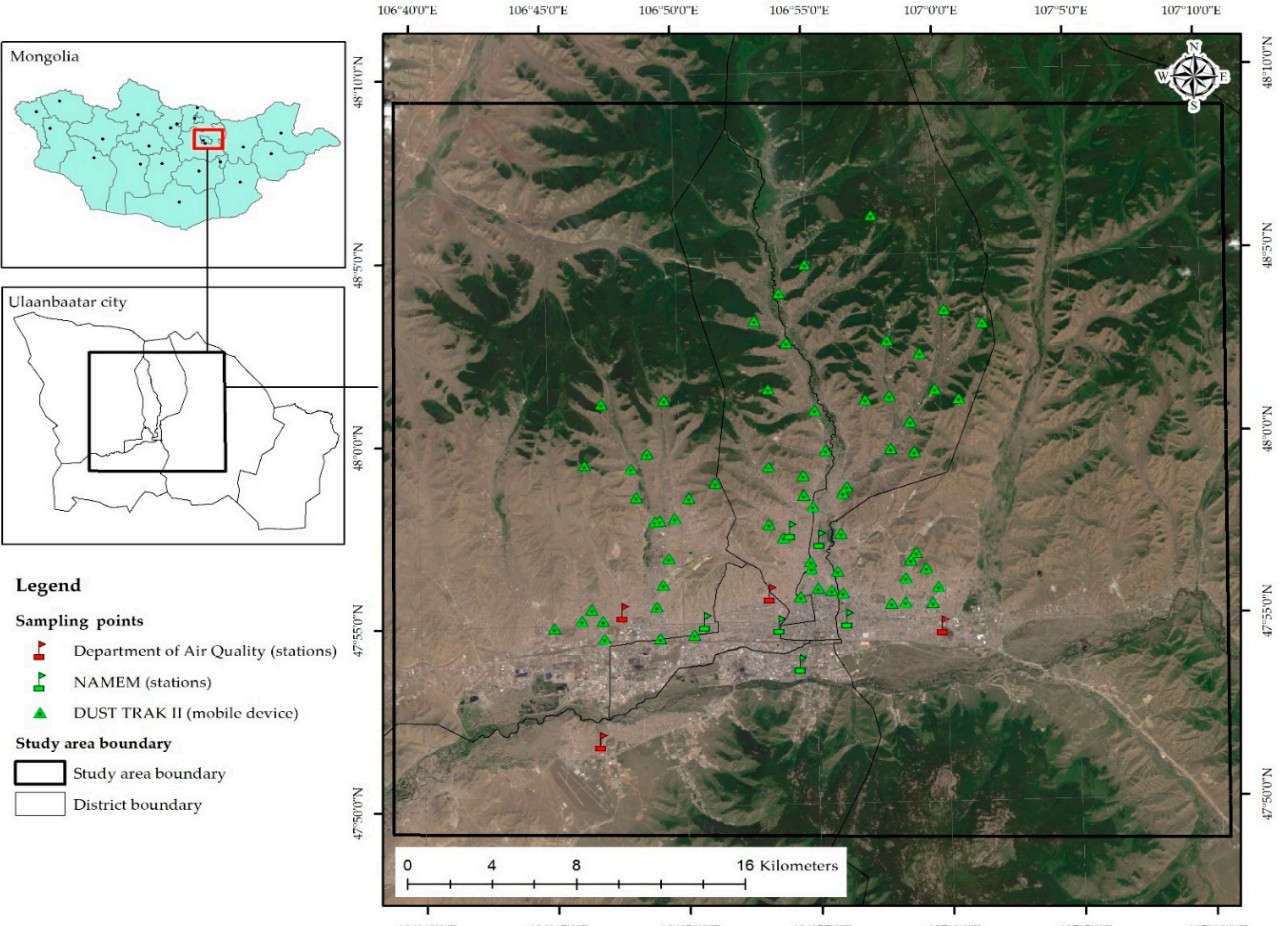

**Figure 1.** Study area and sampling points.

The ENVEA Environnement SA MP101 m Outdoor Air Quality Monitoring Station is a fully automated system for detecting the concentration of airborne particles. In the event that regulatory thresholds are exceeded, this device will sound an alarm. It can monitor particulate matter ($PM_{10}$ and $PM_{2.5}$) continuously. Furthermore, this gauge uses a beta-ray attenuation measurement approach. By measuring the amount of radiation absorbed by a sample collected on a fiber tape after being exposed to a radioactive source, the concentration of particulate matter can be calculated. According to the monitoring stations' measurements on 17 January 2021 (Figure 2), Ulaanbaatar's $PM_{2.5}$ pollution reaches its minimum level in the afternoon, or from 02:00 pm to 05:00 pm. From 06:00 pm until the following morning, it increases continuously and reaches 400 µg/m$^3$. However, these measurements are relatively low due to the stations being located in the city center, far from the ger area. Concentrations measured in the center are considerably lower than those measured in the ger area [20]. The ger area's fine particulate matter pollution is around two times higher than the city center's fine particulate matter pollution [25].

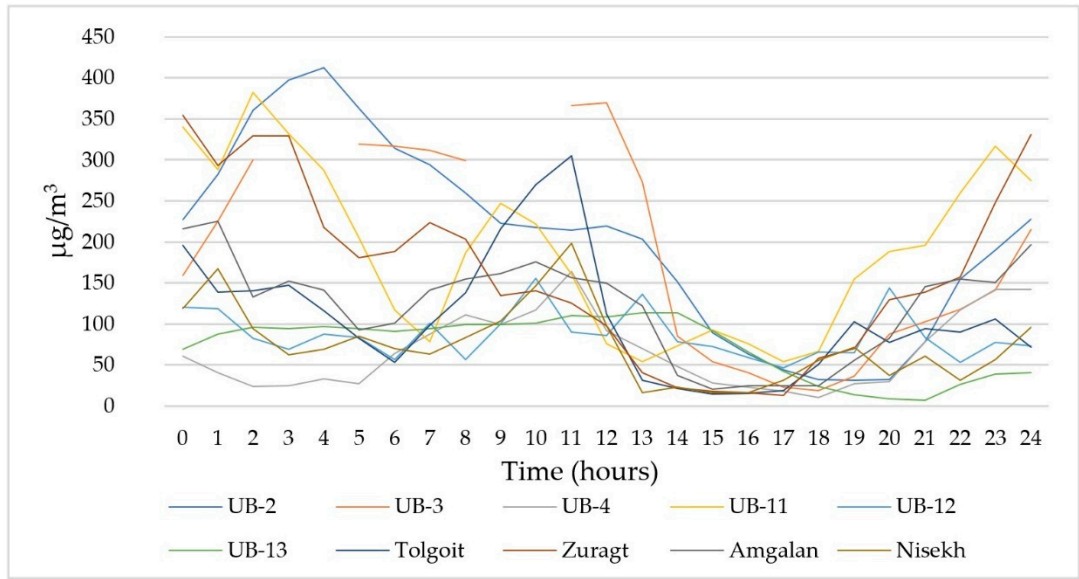

**Figure 2.** $PM_{2.5}$ concentration levels during a day.

Based on other researchers' experience [20–22,26], the "DUST TRAK II Aerosol Monitor 8532" mobile device was borrowed from the Department of Environment and Forest Engineering, National University of Mongolia, and was utilized for the field study. This portable device has a "Certificate of Calibration and Testing" (certificate serial number: 8532134301) and can measure aerosol concentrations for $PM_1$, $PM_{2.5}$, respirable, or $PM_{10}$ size fractions with a corresponding impactor kit. It can also be used to evaluate the interior and outdoor air pollution. The major goal was to completely map the $PM_{2.5}$ pollution in Ulaanbaatar. Sampling was done during the most polluted period of the day, 05:00–11:00 am and 06:00–00:00 pm, on Sunday 17 January 2021, mostly sampling the ger area region. Based on the work of Ryan et al. [20] and Grazio Fattoruso [27], this sample campaign took place over the course of a single day. In addition, the results of the measurements were compared with long-term measurements taken at fixed stations, and since the results of the measurements were found to be consistent with the long-term average measurements taken at fixed stations, the authors concluded that the data could be used to create LUR models. A total of 64 locations were sampled and a total of 74 locations' samples (adding in 10 monitoring stations' samples) were used as dependent variables for the LUR model. Fixed stations and mobile device measurements' descriptive statistics are shown in Table 1.

**Table 1.** Descriptive statistics for measurement data.

|  | Count | Minimum | Mean | Maximum | Median | SD | SE |
|---|---|---|---|---|---|---|---|
| Fixed stations | 10 | 110 | 283.6 | 385 | 305 | 79.76 | 25.22 |
| Mobile device | 64 | 10 | 153.9375 | 405 | 98 | 134.48 | 16.81 |

*2.2. Predictor Variables*

The LUR model is based on the environmental characteristics of the study area, especially characteristics that influence the pollutant emission intensity and dispersion efficiency [28]. Thus, in this study, two types of predictor variables were collected, air pollution sources and the environmental characteristics that influence air pollution dispersion. The integration of road networks, traffic count information, topography, and land cover data within a geographic information system are common throughout all of these models [29]. According to these research results and conclusions, 22 independent variables (Table 1) that were related to Ulaanbaatar's $PM_{2.5}$ pollution were collected. The main sources of $PM_{2.5}$ pollutant (gers and houses) were drawn from Google Earth, according to Weiran Yuchi et al.'s research [1].

Heat-only boiler data were taken from the Department of Air Quality, Ulaanbaatar, all types of road data were obtained from Open Street Maps [30], and ALOS PALSAR Digital Elevation Data were obtained from the Alaska Satellite Facility, US [31]. Landcover data were classified from SENTINEL 2 data. Even though Mongolia has landcover change data dating back 5 years [32], its classification is not sufficient at the city level, so Ulaanbaatar's land surface was classified into 5 types generally. After collecting all data, one full model was built using all these data. However, the results did not show the true dispersion of $PM_{2.5}$. Thus, in order to remove some statistically insignificant independent variables, the Variance Inflation Factor (Table 2) was calculated and the correlation between independent variables was detected. Within the results of these calculations, as for the Variance Inflation Factor, 1 < VIF < 10 levels are generally acceptable for the model [33], and if it is greater than 10, it may create multicollinearity. From these results, VIFs greater than 10 were removed and there was no significant multicollinearity that needed to be corrected [34]. Thus, to remove insignificant variables based on a statistical assessment, the stepwise selection (both forward and backward) method was used. Forward and backward selections are combined in stepwise selection. Researchers begin without any predictor factors and gradually introduce the most beneficial ones (as in forward stepwise selection). Then, we eliminate any predictor variables (such as backward stepwise selection) that no longer improve the model fit [35]. It was then analyzed using the statistical program R, which produced the final model.

**Table 2.** All collected data as independent variables, related to $PM_{2.5}$ pollution in Ulaanbaatar.

| Predictor Variable Type | Predictor Variable | Data Unit | Data Source | Direction | VIF |
|---|---|---|---|---|---|
| Air pollution sources | Gers | density | Google Earth | + | 7.92 |
| | Houses | density | | + | 4.79 |
| | Heat-only boilers | density | Department of Air Quality, Ulaanbaatar City Munincipal | + | 4.03 |
| | Main paved roads | density | | + | 6.75 |
| | Secondary paved roads | density | Open Street Maps | + | 22.34 |
| | Soil roads | density | | + | 10.01 |

**Table 2.** *Cont.*

| Predictor Variable Type | Predictor Variable | Data Unit | Data Source | Direction | VIF |
|---|---|---|---|---|---|
| Environmental characteristics (landcover classification) | Ger area | 1 × 1 pixel buffer | SENTINEL 2 data | + | 168.03 |
| | | 25 × 25 pixel buffer | | + | 599.46 |
| | | 50 × 50 pixel buffer | | + | 198.54 |
| | Wet area | 1 × 1 pixel buffer | | − | 29.19 |
| | | 25 × 25 pixel buffer | | − | 314.01 |
| | | 50 × 50 pixel buffer | | − | 204.18 |
| | Industry area | 1 × 1 pixel buffer | | + | 147.03 |
| | | 25 × 25 pixel buffer | | + | 593.27 |
| | | 50 × 50 pixel buffer | | + | 376.56 |
| | Apartment area | 1 × 1 pixel buffer | | − | 104.85 |
| | | 25 × 25 pixel buffer | | − | 971.73 |
| | | 50 × 50 pixel buffer | | − | 694.10 |
| | Agricultural area | 1 × 1 pixel buffer | | − | 108.35 |
| | | 25 × 25 pixel buffer | | − | 412.35 |
| | | 50 × 50 pixel buffer | | − | 201.44 |
| Elevation | Altitude | above sea level | ALOS PALSAR DEM | − | 7.25 |

### 2.3. LUR Model Development

This research work aimed to map $PM_{2.5}$ pollution during peak hours in Ulaanbaatar. According to the review study by Xing Zhe et al. [23], LUR modeling is the most popular model used in pollution estimation from mobile data and monitoring stations. LUR is based on the principle that pollutant concentrations at any location depend on the environmental characteristics of the surrounding area—particularly those that influence or reflect emission intensity and dispersion efficiency [36]. LUR models were originally developed to assess the exposure resulting from air pollution as a result of vehicular traffic, but they have since been expanded to cover air pollution epidemiology [37]. Recently, this model has been utilized widely, from city-level to continent-level air pollution [38–41].

The LUR model uses linear and nonlinear regression equations with measurement values (Y) and geographical predictor variables as independent variables (X). In this research, two LUR models were developed to model the spatial variability of $PM_{2.5}$ concentrations and produce high-resolution maps with 100 m × 100 m resolution in Ulaanbaatar for the year 2019 (Figure 3).

The Multiple Linear Regression (MLR) equations were used as a linear regression model and the Generalized Additive Model (GAM) was used as a nonlinear multiple regression model. Linear regression models are easy to understand and interpret, used for inference, and allow the understanding of the linear relationship between the dependent and independent variables, but they can suffer from high bias. In other words, machine learning models such as "Gradient Boosting" and "Random Forest" are very useful in making predictions of complex relationships, but they tend to need huge amounts of data and are not easy to interpret. However, the GAM addresses this problem by fitting complex nonlinear relationships and making better predictions. Simultaneously, the GAM allows us to perform inferential statistics, and understand and explain the underlying structure of our used model [42]. As mentioned above in Section 2.2, the final LUR model's independent variables were selected by the VIF and stepwise selection methods.

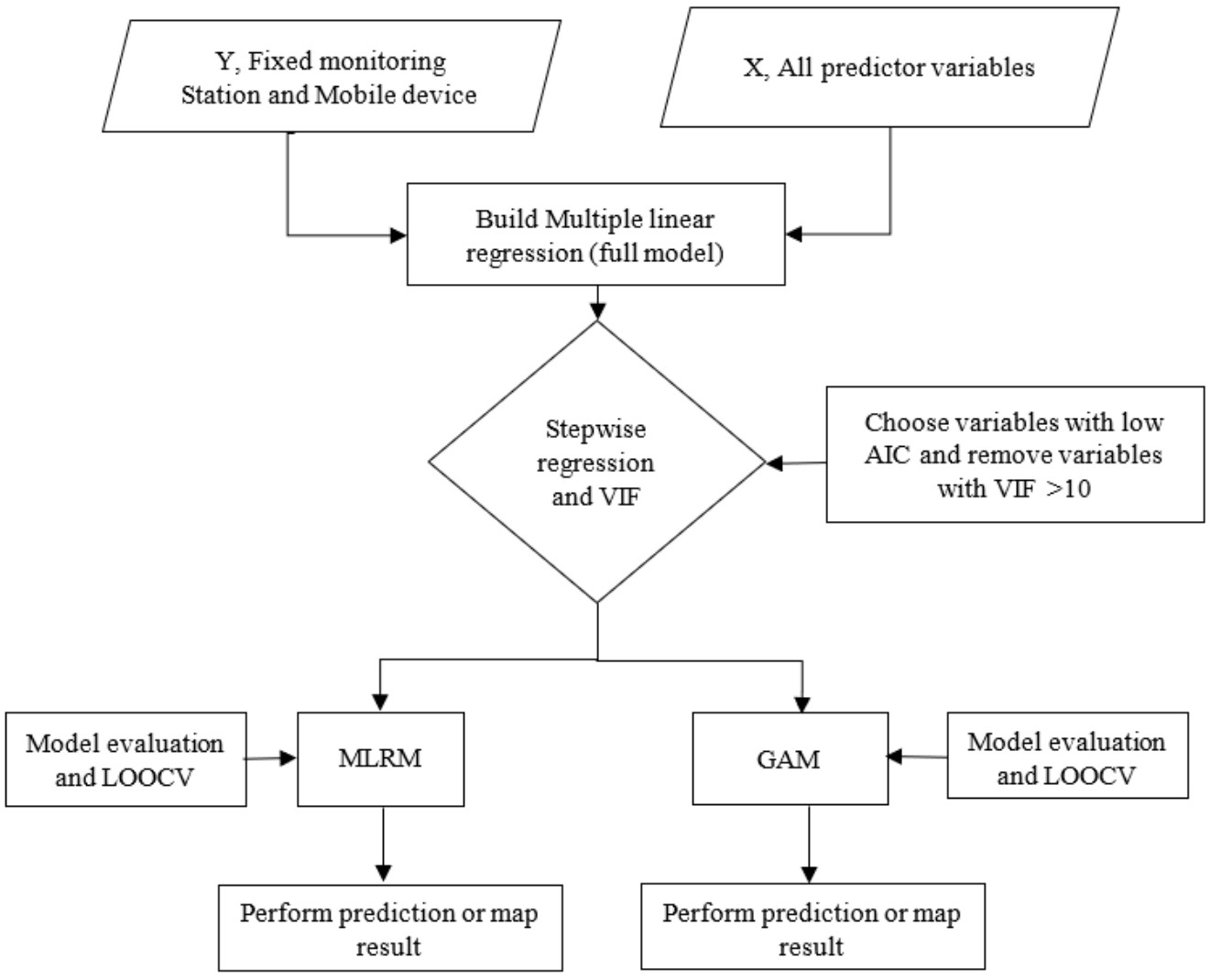

**Figure 3.** LUR model development flow chart.

i.    **MLRM**

$$PM_{2.5} \sim \beta_0 + \beta_1(House) + \beta_2(Ger) + \beta_3(MainPavedRoad) + \beta_4(HeatOnlyBoilers) + \beta_5(AgriculturalArea, 50 \times 50\ buffer) + \varepsilon \quad (1)$$

In Equation (1), $\beta_0$ is the intercept; $\beta_1$ to $\beta_5$ are the $\beta$ coefficients (slopes) of the independent variables; and $\varepsilon$ is the residual (difference between predicted and observed values). $\beta$ coefficients or slopes compare the strength of the effect of each individual independent variable to the dependent variable. The higher the absolute value of the $\beta$ coefficient, the stronger the effect. This means that the variables can be easily compared to each other [43]. $\beta$ coefficients are constant and these coefficients are found with the least square method.

ii.    **GAM**

$$PM_{2.5} \sim \alpha + s(House) + s(Ger) + s(MainPavedRoad) + s(HeatOnlyBoilers) + s(AgriculturalArea, 50 \times 50\ buffer) + \varepsilon \quad (2)$$

In Equation (2), $\alpha$ is the intercept; "s" is the smoothing function of the covariates; and $\varepsilon$ is the error term or the residual. Smoothing functions are specified in a GAM formula using "s", "te", "ti", and "t2" terms. In this study, the default "s" function has been used and it uses thin plate regression splines. These are low-rank isotropic smoothers of any number of covariates. Isotropic means that the rotation of the covariate coordinate system will not change the result of smoothing. "Low-rank" means that they have far fewer coefficients than there are data to smooth. They are reduced rank versions of the thin plate

splines and use the thin plate spline penalty. They are the default smoothers for s terms because there is a defined sense in which they are the optimal smoother of any given basis dimension/rank [44].

### 2.4. Model Validation

In statistics, model validation tests the goodness of fit of the fitted model. Thus, in this part, we used the Leave-One-Out Cross-Validation technique with the R software on both the MRLM and GAM models. In total, 74 samples were used to build 74 models. Each model used 5 predictor variables. No pre-processing occurred and the resampling method was the Leave-One-Out Cross-Validation.

### 2.5. Mapping MLRM and GAM Model

In this research, two statistical and mapping software programs were used, generally ArcGIS 10.4 and R4.2.2. MRLM and GAM models were built and validated in the R statistical software with the "mgcv", "MASS", "ggplot2", and "car" packages, etc. After building the MLRM and GAM models with R, all coefficients and prediction values were exported to the ArcGIS software and calculated with "field calculation".

When spatially aggregated data are used, the modifiable areal unit problem (MAUP) is encountered. Thus, spatial grid points at different scale sizes, such as 100 m × 100 m, 200 m × 200 m, 500 m × 500 m, and 1 km × 1 km, were calculated in order to analyze MAUP sensitivity. There are many methods to perform analysis on MAUP, and spatial autocorrelation is one of them [45]. The Global Moran's I spatial autocorrelation index is a widely used method to analyze MAUP, and, according to this index, for 100 m × 100 m scale aggregation, the Moran's I index is 0.948, and it is the highest clustering index compared to other scale size experiments. In total, 146,211 points across the study region, with 100 m between each point, were used. It is convenient to use the interpolation approach to generate a surface map when working with a grid of equally spaced points because each point's value may be extracted from the raster image as an individual value.

Models were then run using these point-independent values, making it possible to generate a high-resolution map using Equation (3).

$$Z_p = \frac{\sum_{i=1}^{n}\left(\frac{z_i}{d_i^p}\right)}{\sum_{i=1}^{n}\left(\frac{1}{d_i^p}\right)} \tag{3}$$

$Z_p$ = value to be estimated;
$Z_i$ = known value;
$d_i^p$ = distance from the *n* data point to the power *p* of the point estimated.

## 3. Results

### 3.1. PM$_{2.5}$ MLRM and GAM Models

Ulaanbaatar's gers, houses, main paved roads, heat-only boilers, and agricultural land were used as independent variables in the final linear LUR model constructed using the stepwise selection results, with the dependent variable (Y) originating from PM$_{2.5}$ measurements taken at fixed stations and mobile devices. The final linear model was defined as follows:

$$Y = 1.33259 + 15.85699 * \text{ger} + 5.28564 * baishin + 0.23285 * \text{r\_m\_p} + 2.00580 * \text{stoves} + 0.04753 * \text{F50\_agri}$$

Table 3 shows detailed information about the MLRM model, including the constant coefficients, standard error, *t*-value, *p*-value, and Variance Inflation Factor. The standard error of all independent values ranged between 0.006 and 4.02, and the same was true for the beta coefficient. The t-value was also between 0.117 and 6.953, or far from "0", and we could thus reject the null hypothesis and prove that there were relationships between

the dependent and independent variables. As for the *p*-value (Pr(>|t|)), gers, main paved roads, and agricultural land have high significance, while other independent values offer insufficient evidence against the null hypothesis. Moreover, all VIF coefficients are less than 5, meaning that there is no high correlation between the independent variables.

**Table 3.** Detailed information of the MLRM.

| Independent Variable | Code Name | Estimate | Std. Error | *t*-Value | Pr(>|t|) | Variance Inflation Factor |
|---|---|---|---|---|---|---|
| α | Intercept | 1.33259 | 11.426472 | 0.117 | 0.907502 | - |
| Gers | ger | 15.85699 | 4.029674 | 3.935 | 0.000198 | 3.17 |
| Houses | baishin | 5.28564 | 3.801982 | 1.390 | 0.168992 | 3.21 |
| Main paved roads | r_m_p | 0.23285 | 0.065516 | 3.554 | 0.000695 | 1.77 |
| Heat-only boilers | stoves | 2.00580 | 1.068806 | 1.877 | 0.064855 | 2.51 |
| Agricultural land | F50_agri | 0.04753 | 0.006836 | 6.953 | $1.72 \times 10^{-9}$ | 1.68 |

After building the MLRM model, all independent variables were also utilized for the nonlinear regression GAM. The effective degrees of freedom (edf) estimated from the GAM model were used as a proxy for the degree of nonlinearity in stressor–response relationships. According to Table 3, the edfs of independent variables such as houses, main paved roads, and heat-only boilers are equal to 1 and they have a linear relationship; for other independent variables, such as gers and agricultural land, the edfs are more than 2 and this indicates that they have a highly nonlinear relationship [46]. The *p*-value shows the significance level of independent variables. Gers and agricultural land's values are less than 0.05, or they have significance, while others are more than 0.05, or they offer insufficient evidence against the null hypothesis (Table 4).

**Table 4.** Detailed information of the GAM.

| Independent Variable | Code Name | edf | *p*-Value |
|---|---|---|---|
| Gers | ger | 4.621 | 0.0168 |
| Houses | baishin | 1.000 | 0.6364 |
| Main paved roads | r_m_p | 1.000 | 0.1117 |
| Heat-only boilers | stoves | 1.000 | 0.1150 |
| Agricultural land | F50_agri | 5.431 | $<2 \times 10^{-16}$ |

### 3.2. Model Accuracy and Validation

Several statistical metrics were calculated for the model assessment for both models (MLRM and GAM) and LOOCV. Statistical metrics used in this study were the determination coefficient ($R^2$), root mean square error (RMSE), adjusted $R^2$, and mean absolute error (MAE), which are widely used for model assessment. Figure 4 and Table 5 (LOOCV column) present the plots of the predicted and observed values, model accuracy, and LOOCV validation coefficients. As for the accuracy of the MLRM model, determination coefficient $R^2 = 0.84$, adjusted $R^2 = 0.83$, and RMSE = 53.25 µg/m$^3$, and *p*-values were less than the significance level. As for the accuracy of the GAM model, the accuracy was slightly higher than that of the MLRM, or $R^2 = 0.89$, adjusted $R^2 = 0.87$, and RMSE = 44 µg/m$^3$. The *p*-values of these models were $2 \times 10^{-16}$ or much less than the significance level of 0.05, and the null hypothesis was rejected. Based on these assessment coefficients, the MLRM and GAM models predict reasonably well.

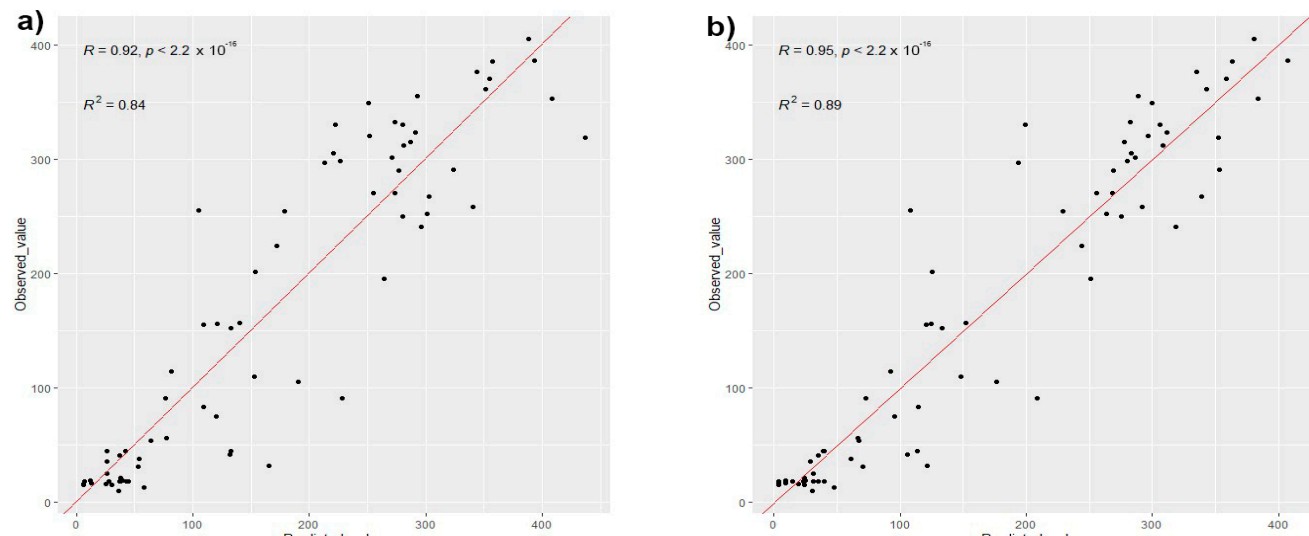

**Figure 4.** Observed and predicted value relationship for the (**a**) MLRM and (**b**) GAM.

**Table 5.** Model performance accuracy and validation.

| Model Type | Fitted Model | | | | LOOCV | | |
|------------|------|------|-------------|---------|------|------|------|
| | $R^2$ | RMSE | Adjusted $R^2$ | *p*-Value | $R^2$ | RMSE | MAE |
| MLRM | 0.84 | 53.25 | 0.83 | $2.2 \times 10^{-16}$ | 0.83 | 55.6 | 38.7 |
| GAM | 0.89 | 44.0 | 0.87 | $2.2 \times 10^{-16}$ | 0.77 | 65.5 | 47.7 |

As for model validation, validation technique LOOCV was used and its accuracy coefficients were similar to the model fit assessment coefficients (Table 4): MLRM LOOCV determination coefficient $R^2$ = 0.83, RMSE = 55.6 µg/m$^3$, and MAE = 38.7 µg/m$^3$; GAM LOOCV determination coefficient R2 = 65.5 µg/m$^3$, RMSE = 65.5 µg/m$^3$, and MAE = 47.7 µg/m$^3$. Furthermore, there was no large difference or error between the fit model and LOOCV model.

*3.3. Mapping*

The MLRM and GAM models were run on each point within 146,211 grid points, with 100 m distances between them. Values of each point were interpolated with the Inverse Distance Weighting (IDW) method (Formula (3)), and a surface map was created (Figure 5). These maps depict the amount of PM$_{2.5}$ pollution on 17 January 2021, one of the coldest days of the winter season. In other words, during the winter months, when most people stay at home and burn coal to heat their homes, for the peak hours of the day, such as early in the morning and late at night, there are significant levels of air pollution. In order to show exposure to PM$_{2.5}$, the EPA's Air Quality Index for 24-h Fine Particle Pollution Levels [47] was used for the legend. According to its categories, 0–12 µg/m$^3$—"good", 12.1–35.4 µg/m$^3$—"moderate", 35.5–55.4 µg/m$^3$—"unhealthy for sensitive people", 55–150.4 µg/m$^3$—"unhealthy", 150.5–250.4 µg/m$^3$—"very unhealthy", <250.5 µg/m$^3$ —hazardous. From this, Ulaanbaatar's PM$_{2.5}$ pollution exceeds the hazardous level and this situation continues for nearly every half day in the winter months. Additionally, Ulaanbaatar's topography, being surrounded by mountains, blocks air movement and increases the accumulation of pollution throughout the city.

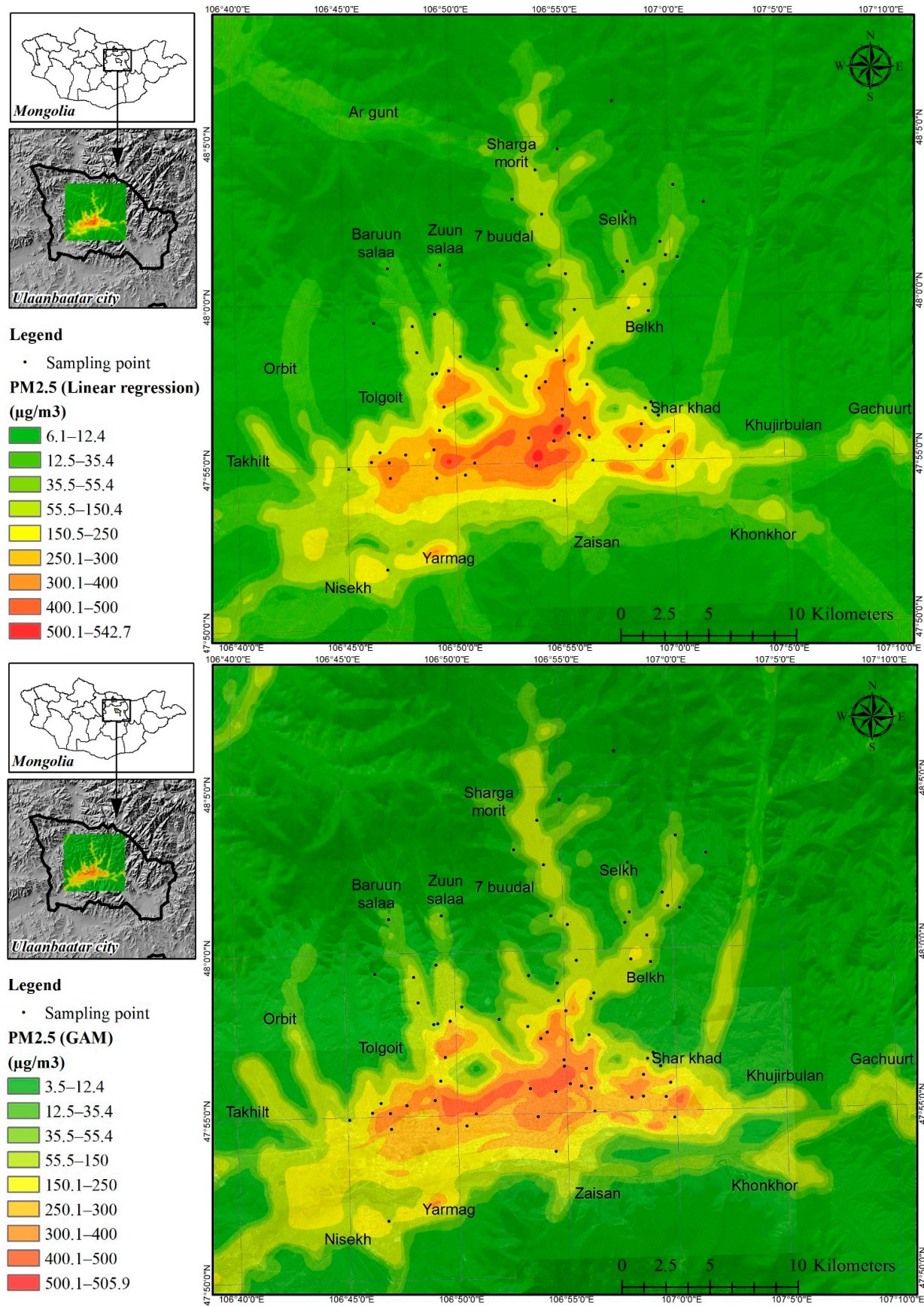

**Figure 5.** MLRM and GAM result map.

Finally, the PM$_{2.5}$ dispersion maps show a spatial dispersion intensity in Ulaanbaatar City similar to the results of Yukimasa et al. [12], Ryan W.Allen et al. [1], and the "Project to strengthen air pollution control capacity in Ulaanbaatar, Mongolia" (PSAPCCU) report [48]. Yukimasa et al. used the diffusion simulation method, but its results show the total emissions of 6 full hours of combustion with each stove's particulate matter emission capability. Ryan W. Allen et al. used the LUR model, but they showed the spatial variability of the annual average PM$_{2.5}$ concentration and did not report model prediction errors. The PSAPCCU mapped air pollution dispersion with the CALMET + CALPUFF model. Its spatial variability was determined as grid cells, and compared to this research, it is inaccurate. We did not find similar research using LUR modeling for PM$_{2.5}$ over Ulaanbaatar City. The abovementioned results, despite using different methods, show similar spatial variability in the dispersion result. The main advantage of this study's results compared to previous studies is more detail and wider coverage of the residential areas of Ulaanbaatar.

## 4. Discussion

A few studies have attempted to model particulate matter pollution exposure at the city level in Ulaanbaatar [1,4,12,21,22]. However, the resulting maps are not highly detailed and are not very clear. Weiran Yuchi et al. [1] tried to map air pollution dispersion with the LUR model using independent variables such as wetness, greenness, gers, and road lengths. However, they did not create a map, showing only the independent variable dispersion map. According to their results, independent variables explained an average of 70% of the model. This research could not issue a satisfactory result. Sarath Guttikunda et al. [4,49] mapped air pollution with the ATMOS model, but their results' spatial variability was not accurate. Yukimasa et al. [12] and Ryan W et al. [20] also tried to map air pollution with statistical data, and their result regarding spatial variability was similar to the results of this study, but their spatial accuracy was too poor. In recent years, the Department of Capital City Air Quality has measured particulate matter, but its stations are mainly located downtown, and measurement results are issued on the website with dot marks. Thus, the accurate modeling of particulate matter pollution is still a priority issue in Ulaanbaatar.

In this research, data relevant to Ulaanbaatar's PM$_{2.5}$ pollution were collected as independent variables for the LUR model. Many of them were not well suited to the LUR model due to the characterization of Ulaanbaatar's air pollution sources. For instance, many researchers who have developed a LUR model used altitude or a Digital Elevation Model (DEM) as an independent variable. This contrasts with Ulaanbaatar's ger area, which is the main source of PM$_{2.5}$, being mainly located on the tops of mountains. If a DEM was used in this LUR model, it would show an incorrect prediction suggesting that areas with higher elevation have high pollution. Thus, DEM data are not useful for the Ulaanbaatar LUR model because they would show that higher places have more pollution and the residential areas located in lower places have less pollution.

Another finding was that many researchers use population density data for an LUR model based on the principle that higher population density leads to higher air pollution. This is contrary to Ulaanbaatar, where ger area residents mainly live in one-floor houses or "ger" dwellings as a single family within a fenced area of around 0.07 hectares. Apartment residents live in multi-floored apartments, having many families in a small area. In other words, in the ger area, every single family emits PM$_{2.5}$ pollution through a chimney when burning coal, and apartment-residing families do not, since they are connected to the central heating system. This would show an incorrect prediction that low-population-density areas generate high pollution. Thus, population density data are also not useful for the Ulaanbaatar LUR model.

European researchers mostly use CORINE land cover monitoring data [50] for an LUR model. This integrated land cover database shows integrated results for LUR model development. As for Ulaanbaatar, there are no land cover monitoring data at the city scale. Thus, in this study, the researchers classified land cover themselves, and this will likely result in different study results as compared with other similar studies. Even though

Mongolia has land cover monitoring data at 5-year intervals, they are not suitable at the scale needed for the analysis of the city. There is a requirement to monitor city-level land cover change, as with CORINE land cover data. This type of data will be fundamental, resulting in integrated data not only for air pollution studies but also various other environmental studies. Consequently, in this study, only agricultural land cover data were utilized for the model as an independent variable.

Ulaanbaatar is situated in the Tuul river valley, surrounded by high mountains. This topographic condition is the main factor in blocking air movement and increasing the accumulation of air pollution [51]. Additionally, this is one factor that creates "an inversion layer" over the city. This inversion layer appears as black smoke clouds over the city, and the accumulated clouds easily differ from cleaner air. When $PM_{2.5}$ was sampled at the top of the mountains, the city appeared as if it was diving into smoke. This characteristic was detected in the resulting image of this study from the drastically differentiated $PM_{2.5}$ levels in the mountain slope regions.

This research's main findings include the more detailed spatial variability of $PM_{2.5}$ pollution and greater coverage of the residential areas of Ulaanbaatar City compared to previous studies. In addition, the results of this research were evaluated with some statistical methods, and the evaluation coefficients were reasonably high. This research's results are very accurate and can have very practical uses. For instance, the Mongolian government is planning to build "micro centers", which are infrastructures that aim to address the most polluted "ger" areas within the scope of the National Program to Decrease Air and Environmental Pollution [52]. These "micro centers" are planned to include 200–300 householders and will be provided with a heating system. It is hoped that this research can offer a recommendation in choosing where to build "micro centers" in the ger area.

*Limitations*

Our LUR models combined the data from a mobile device and a fixed station into a single model due to the fact that there were only 10 fixed stations, the majority of which were downtown. These measurement data could not cover all residential areas, and thus we could not build the models properly. Thus, there is a need to measure air pollution on the outskirts of the city or in the ger area. We borrowed a DUST TRAK II mobile device for $PM_{2.5}$ measurement from the Department of Environment and Forest Engineering, National University of Mongolia, and performed a short-term sampling campaign with this device based on the experience of Ryan et al. [20] and Grazio Fattoruso [27]. In our study, in order to show the highest pollution concentration, measurement was conducted early in the morning and in the evening. Thus, it is impossible to compare this result with daytime air pollution.

Our model might be constrained by the availability of predictor data. Gers, homes, and heat-only boilers in Ulaanbaatar are the main sources of $PM_{2.5}$ pollution, since they employ coal combustion stoves. Even though heat-only boilers and gers and houses are the leading sources of $PM_{2.5}$ pollution in Ulaanbaatar, the Department of Capital City Air Quality only keeps track of heat-only boiler statistics. Since gers and houses on the outskirts of the city must be included in models as the primary source of $PM_{2.5}$ pollution, the authors made the decision to collect data from Google Maps in response to the study by Weiran Yu Chi et al. Google Maps data, on the other hand, remained largely stable over the course of the three years, indicating that the final two models are largely accurate in reflecting the spatial variability in $PM_{2.5}$ pollution throughout the year 2021.

## 5. Conclusions

The aim of this study was to show the highest $PM_{2.5}$ pollution dispersion in Ulaanbaatar by measuring pollution during peak hours, and to detect the most polluted areas of the city. The linear (MLRM) and nonlinear (GAM) LUR models, a widely used approach to assess exposure to air pollution, were used as the primary method for dispersion mapping and their performance was high, with MLRM $R^2 = 0.84$, adjusted $R^2 = 0.83$,

RMSE = 53.25 µg/m$^3$, and GAM R$^2$ = 0.89, adjusted R$^2$ = 0.87, RMSE = 44 µg/m$^3$. In order to validate the models, the LOOCV technique was run on both MLRM and GAM. Their performance were again high, with MRLM LOOCV R$^2$ = 0.83, RMSE = 55.6 µg/m$^3$, MAE = 38.7 µg/m$^3$, and GAM LOOCV R$^2$ = 0.77, RMSE = 65.5 µg/m$^3$, MAE = 47.7 µg/m$^3$. From these results, GAM works better than MRLM or its fit model, and the LOOCV assessment coefficients are higher than those for the MRLM fit model and LOOCV.

Sources contributing to PM$_{2.5}$ were dominated by emissions from coal combustion in the ger area. Most householders in the ger area use coal-burning stoves as a home heating device. Furthermore, due to heating of the home and cooking, both in the morning and evening, PM$_{2.5}$ pollution increases heavily, and exceeds the "hazardous" level. According to this study's results, Ulaanbaatar's PM$_{2.5}$ pollution reaches over 500 µg/m$^3$. This occurs mainly in the city center during the winter, early in the morning, and in the evening, continuing for around half a day. In addition to excessive smoke emissions, the topography of the capital city plays a significant role. In the Tuul river valley, surrounded by mountains, an inversion layer is created, normal air mixing ceases, and pollutants are trapped in the lower layer. As a result of this, PM$_{2.5}$ pollution accumulates significantly, especially on calm days, where it can be 2–3 times higher than the hazardous level.

With such high fine particulate matter concentrations, there are likely to be significant health implications for Ulaanbaatar's citizens. Based on the findings of this study, stoves used for heating are the primary cause of PM$_{2.5}$ pollution. Therefore, there is a need to switch from stoves to various fuel-free heating options. In other words, the ger region needs to adopt electrical heating gadgets or a central heating system. This will reduce PM$_{2.5}$ pollution more successfully. Additionally, the resulting map shows the intensity of the level of air pollution, and people should avoid living in and visiting areas with high levels of pollution.

During the experimentation phase of model construction, our LUR models revealed that altitude and population density data are not always adequate due to PM$_{2.5}$ pollution source dispersion, which is a theoretical contribution to the LUR model. The ger district of Ulaanbaatar, the city's primary contributor to PM$_{2.5}$ pollution, is located in relatively remote, high-altitude areas; its population is dispersed sparsely compared to the downtown, densely populated, central district. In contrast, according to the laws of physics, particulate matter pollution must build up in lower places, and places with a large number of people must support it heavily. In Ulaanbaatar, caused by particulate matter pollution source dispersion, this pattern is the opposite.

Finally, these maps illustrate the extent to which air pollution is a problem and how much of a geographic area is affected. We anticipate that these data will be useful in helping decision-makers to identify areas where urgent action is required to reduce air pollution.

**Author Contributions:** Conceptualization, X.-Y.Y. and O.E.; methodology, O.E.; software, O.E.; validation, O.E.; formal analysis, O.E.; investigation, O.E.; resources, O.E.; data curation, O.E.; writing—original draft preparation, J.C.; writing—review and editing, M.L.; visualization, O.E.; supervision, X.-Y.Y.; project administration, O.E.; funding acquisition, M.L. All authors have read and agreed to the published version of the manuscript.

**Funding:** This research was funded by a National Research Foundation of Korea (NRF) grant provided by the Ministry of Education (No. 2021R1I1A1A01060652).

**Acknowledgments:** We wish to thank Chonokhuu Sonomdagva (Department of Environment and Forest Engineering, National University of Mongolia), who lent us the expensive "DUST TRAK II" mobile device for the field research. We also thank Ganbaatar Ochirbat, and researchers at the Divisio/n of Innovation Technology, Institute of Physical Technology, MAS, who spent a full day with us and helped to sample the PM$_{2.5}$ for field measurement. With their help, the field measurement was performed effectively and efficiently, and this provided the most significant data for this research work.

**Conflicts of Interest:** The authors declare no conflict of interest.

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
