# Peer review of "Linear and Nonlinear Land Use Regression Approach for Modelling PM2.5 Concentration in Ulaanbaatar, Mongolia during Peak Hours"

_remotesensing, doi:10.3390/rs15051174_

Round 1

Reviewer 1 Report

This paper uses a linear and nonlinear land use regression method to model PM2.5 distributions in Ulaanbaatar, a city of Mongolia. The method is not new and the authors argue that their contribution to extant literature lie in a finer PM2.5 modeling in the city of Ulaanbaatar. In my opinion, the novelty of this work is weak and authors solely apply LUR in a new area while the discussion on the case is insufficient. My detailed comments are as follows.

1.       In the introduction, you use lots of texts to introduce the situation of Ulaanbaatar while lacking sufficient review on the research status of recent works especially for the spatial modeling technique of pollutants or PM2.5. Please add relevant contents. As I mentioned before, your work focus on providing a throughout mapping of PM2.5 in a new area. The theoretical contribution to existing knowledge should be further justified.

2.       line 128, mapmore may be a typo. Also, there are some other minor language errors in your paper, please make careful proofreading.

3.       line 158-159, multicollinearity check is unable to detect insignificant variables.  

4.       Please explain the model in more detail in section 2.3 (especially for GAM) and describe how these models are estimate. Now, the method part is unclear.

5.       line 202, why choose 202 as the spatial unit? And, please consider the Modifiable areal unit problem (MAUP) because this is very critical for spatial modeling.

6.       line 206, please number the formula and cite it in the text.

7.       Table 2, please report p-value.

8.       line 225, as the independent variables have different dimensions/units, a large coefficient does not mean more sensitive effect. Thus, why you report that Highest sensitive independent values are “Gers” and “Houses”.

9.       line 237, I see only the coefficients of Gers and Agricultural land are significant.

10.    From the value of RMSE (53.25 and 44), I feel the models do not perform excellent because the value seems a bit large. In fact, we care more about the prediction error instead of R2.

11.    Please provide and discuss more information on the PM2.5 modeling results in the case of Ulaanbaatar.

12.    line 276-279, please consider comparing the prediction error of your work with existing studies in similar areas.

13.    Please discuss the limitation of your research.

Author Response

Dear, 

Very thank you for your comments. I have replied to all comments and coloured with red.

best regards,

Reviewer 2 Report

The manuscript entitled “Linear and Nonlinear Land Use Regression Approach for Modelling PM2.5 concentration in Ulaanbaatar, Mongolia during peak hours” aims to map PM2.5 dispersion using the Linear and Non-linear regression approach. The manuscript nicely gives emphasis on the high PM2.5 concentration in the Mongolian capital city. The manuscript needs enhancements before publication. I suggest authors to address the following comments and improve the manuscript, before publication.

A)    Sampling: How many days of sampling have been carried out and for how long? Line 130 authors are saying they sampled on 17th Jan 2021. Please justify how this small sampling dataset is helpful to derive all the results.

B)    Please improve Figure 2.

C)    Line 141-143: Please explain the Google Earth use to drown PM2.5 sources.

D)    Line 368-369: Not clear --‘This will be a more effective way to decrease PM2.5 pollution.’ What is the most effective way to decrease PM2.5 Pollution?

Authors would better use satellite AOD datasets and relate them to their PM2.5 surface concentration. The manuscript is silent about the possible ways to control the PM2.5 in the city. Further, authors may investigate Urban Heat Island over the city in their future work.

Author Response

Dear,

Very thank you for your comments. I have replied to all comments and coloured with green. And other colours are the denotation of overlapped answer. 

best regards

Reviewer 3 Report

Overall, this manuscript used linear (MLR) and non linear (GAM) Land Use Regression to model spatial PM2.5 concentration in Ulaanbaatar, Mongolia. The scientific value was presented in this manuscript, however, a lot of amendment need to be done for perfecting this manuscript. Below are some major comments:

1. Title - represent the body of the manuscript.

2. Introduction - Lack of literatures discussed on the spatial models that currently used. No discussion on the previous work on the spatial model in Mongolia making me think that this was the first. However, a few studies had been conducted and the authors need to highlight the gap of their study with the previous literature. 

3. Methodology - No details of the monitoring stations and the "dusttrak II" were described. No description on training and validation of data. I suggest the author should include the flow chart of method and explain in details each of the research activities prior to the flowchart. No details description as well on the performance measure and the statistical aspect especially on the hypothesis and the alpha value.

4. Results - It was stated in the title that the prediction of PM2.5 was done during the peak hour. From what I understand, the peak hour of PM2.5 are during early morning and evening, hence I expect at least the modeling will be at these hours. However, only 1 model was presented each for MLR and GAM. Because no clear explanation of the methods, hence create confusion in the results. The authors should describe the statistics of the data from monitoring stations, dusttrak and others before explaning the model.

5. Discussion- The authors tend to discuss only literatures without relating the findings in this paper. Somehow, I think most of the discussion part should be placed in introduction. The authors should discuss the findings on the models and what make this findings beneficial compared to other studies.

6. Conclusion - the authors should add the restrictions and how this research can help for better management of air quality in Mongolia especially during winter.

The details of comments is given in the attached document.

Author Response

Dear,

Very thank you for your comments. I have replied to all comments and coloured with blue. And other colours are the denotation of overlapped answer. 

best regards

Reviewer 4 Report

This article by Enkhjargal et al. is about modelling particulate matter (PM2.5) concentration in the atmosphere by the mean of land use regression (LUR). The authors motivate their study by the degradation of air quality in the traditional "ger" housing areas of Ulaanbaatar (Mongolia). This pollution is due to due to coal combustion for cooking and heating in the morning and evening, which creates a difference between day and night PM2.5 concentration levels. Accurate mapping of air pollution is very important to assess the exposure of the population. LUR is a common method to map small-scale spatial distribution, for instance of air pollutants, based on measurements and geographical predictors. Thus, authors develop a multi-linear regression model (MLRM) and a generalized additive model (GAM) to map peak PM2.5 dispersion during the day. To do this, they make use of measurements from monitoring station and from a mobile device and they validate the models by the leave one out cross validation technique. According to their results, the authors conclude that the LUR is efficient and the GAM works better than the MRLM.

This article deals with the important subject of air quality with a practical application to the Mongolian traditional house quarters of Ulaanbaatar. The authors choose land use regression methods, which is a perfectly acceptable choice. I don't understand why the authors claim that they can't take a physical model of the air quality. I assume that there is very little quantitative and/or sufficiently precise data on pollutant emissions. This point should be clarified. Other than that, the article is written correctly and relatively clear. Although the topic and the method are not very original, the application is of interest. Thus, the article will deserve to be published after the authors have taken into account my comments below.

L 016 - The authors should specify what is a "ger" house.

L 020 - I don't agree with the sentence: "... air pollution stations are too scarce in Ulaanbaatar to develop an air dispersion model". What possibly prevents the use of a physical dispersion model is not the scarcity of measurements, but the lack of qualitative and/or quantitative information on emissions. Please modify (or justify your assertion).

L 119 - Figure 2 - What is the time scale? Is it local time?

L 155 - The auhors should recall what is "multicollinearity".

L 157 - There are many potential predictor variables in Table 1 with VIF greater than 10. Please comment.

L 160 - The authors should briefly recall what is the "stepwise selection" used forward and backward.

L 219 - What does "Y" stand for? Is it the PM2.5 concentration? Please clarify.

L 228 - Words are missing between "relationships" and "dependent and independent variables". Please rephrase.

L 254 - "Predicting highly" is not correct wording. Please rephrase.

L 267 - At what time of day are the maps in Figure 4 drawn? It's not mentioned in the article. This must be explained and the choice of this time justified at this point in the article.

Author Response

Dear,

Very thank you for your comments. I have replied to all comments and coloured with purple. And other colours are the denotation of overlapped answer. 

best regards 

Round 2

Reviewer 1 Report

Thanks for your revision. I have reviewed the updated manuscript and it improves to some extent. However, some responses/modifications do not satisfy my comments before.

In introduction, please reduce the introduction of Ulaanbaatar, some could be moved to Section 2.1.

As I mentioned in my last review, your work focus on providing a thorough mapping of PM2.5 in a new area. The theoretical contribution to existing knowledge should be further justified.

Please again check your language, not only some issues I pointed out before.

As regards the multicollinearity, you revised relevant description as VIFs greater than 10 were removed and there are no correlations between independent variables. This is inappropriate as no multicollinearity does not mean no correlations.

For "why choose 202 as the spatial unit? And, please consider the Modifiable areal unit problem (MAUP) because this is very critical for spatial modeling." The authors do not address my proposed concerns. Please explain the suitability of 202. With this, please also consider MAUP and discuss it. This is very important.

As to "as the independent variables have different dimensions/units, a large coefficient does not mean more sensitive effect. Thus, why you report that Highest sensitive independent values are “Gers” and “Houses”" Your response is "The beta coefficient is the degree of change in the outcome variable for every 1-unit of change in the predictor variable" This is right only if all the variables have the same dimension. If you still want to compare the coefficients across independent variables, please use the standard beta coefficients (namely first normalize all variables and then run OLS)

Author Response

Thank you for your comments. 

Reviewer 3 Report

All issues/ comments has been addressed appropiately.

Author Response

I am very glad for all my replies were appropriate. Very thank you. 
